# Three Successfully Treated Cases of *Lodderomyces elongisporus* Fungemia: Case Reports and a Review of the Literature

**DOI:** 10.3390/microorganisms11041076

**Published:** 2023-04-20

**Authors:** Nobuhiro Asai, Yuichi Shibata, Akiko Nakamura, Hiroyuki Suematsu, Atsuko Yamada, Tomoko Ohno, Daisuke Sakanashi, Yuzuka Kawamoto, Narimi Miyazaki, Isao Koita, Hideo Kato, Mao Hagihara, Hirotoshi Ohta, Hiroshige Mikamo

**Affiliations:** 1Department of Clinical Infectious Diseases, Aichi Medical University Hospital, Nagakute 480-1195, Japan; 2Department of Infection Control and Prevention, Aichi Medical University Hospital, Nagakute 480-1195, Japan; 3Department of Pathology, University of Michigan, Ann Arbor, MI 48109, USA; 4Department of Pharmacy, Mie University Hospital, Tsu 514-0001, Japan; 5Department of Clinical Pharmaceutics, Division of Clinical Medical Science, Mie University Graduate School of Medicine, Tsu 514-0001, Japan; 6Department of Molecular Epidemiology and Biomedical Sciences, Aichi Medical University, Nagakute 480-1195, Japan

**Keywords:** *Loderomyces elongiporus*, matrix-assisted laser desorption/ionization mass spectrometry, *Candida parapsilosis*, fungemia

## Abstract

Fungemia is a fatal systemic infection that can occur in immunocompromised patients. Despite that, antifungal stewardship is spreading widely, but the mortality rate is extremely high, showing 40–60%. *Loderomyces elongiporus* is a newly morphologically detected pathogen, first described in 1994, followed by isolation in humans in 2008. It has been misrecognized as *Candida parapsilosis*. Recently, fever attributable to *L. elongisporus* fungemia cases has been reported, and the etiology and clinical features are still unknown. Here, we present three successfully treated *L. elongisporus* fungemia cases by echinocandin. In total, 11 cases were reviewed, including ours. Six of the eleven cases (55%) had external devices. All cases had some immunocompromised conditions or underlying diseases, such as diabetes mellitus, lung cancer, etc. Six patients survived, and the remaining five died. Seven patients who had received echinocandin initially survived. Risk factors for *L. elongiporus* fungemia overlap with those of candidemia. Even though there is no breakpoint for *L. elongiporus*, echinocandin can be a helpful treatment regimen for *L. elongiporus* fungemia.

## 1. Introduction

With advanced diagnostic methods, such as matrix-assisted laser desorption/ionization mass spectrometry (MALDI-TOF MS), newly detected pathogens have been increasing year by year [1,2]. In addition, various immunosuppressive agents cause the emergence of opportunistic infections. In spite of a global spread in antifungal stewardship, invasive fungal infection such as catheter-associated blood stream infection caused candidemia, still results in critical severe cases that show a high mortality rate of 46–75% [3,4,5].

“*Lodderomyces elongisporus* was first recovered from Californian citrus concentrate in 1952 and named *Saccharomyces elongisporus* by Recca and Mrak. It was later renamed as *Lodderomyces elongisporus* by van der Walt and colleagues in 1966 [6]. *L. elongisporus* was considered a teleomorph of *Candida parapsilosis* until sequencing of the 18S rRNA gene differentiated it as a unique species. While it has been recovered from fresh fruit, fruit concentrates, and soft drinks, *L. elongisporus* has also been recovered from insects [6]”. *L. elongiporus* is an uncommon yeast opportunistic pathogen that can cause a sporadic fungal infection. *L. elongisporus* is an uncommon yeast opportunistic pathogen that can cause a sporadic fungal infection. This fungus was newly morphologically identified in 1994 [7], followed by its isolation in 2008, as a causative pathogen of human infection [8]. While several fungemia cases caused by this fungus have been reported [6,9,10,11,12,13,14,15,16], we still know little about its epidemiology, clinical features, risk or prognostic factors. Here, we present three cases of catheter-related bloodstream infection (CRBSI)s caused by *L. elongisporus*, which were successfully treated. Furthermore, we review all known reports published previously of fungemia caused by *L. elongisporus*.

## 2. Case Presentation

After a cardiac catheter test, a 76-year-old man was admitted to our institute due to an infectious aneurysm in the right inguinal artery. He had a history of poorly controlled diabetes mellitus with hemoglobin A1c 8.6% and angina pectoris. Since the causative pathogen for the disease was *Staphylococcus aureus*, he received combination antibiotic therapy with sulbactam/ampicillin and daptomycin for 4 weeks. Then, his condition was improved once. During the stay, he had a high fever, and two sets of blood cultures revealed yeast in both aerobic and anaerobic samples (BD, Tokyo, Japan). His central venous catheter (CVC) was removed, and empiric antifungal therapy with micafungin (MCFG) was started simultaneously. The yeast was identified as Lodderomyces elongisporus by MALDI-TOF MS using the VITEK-2 system (BioMérieux, Marcy l’Etoile, France). Since the tip of the CVC also revealed *L. elongisporus*, he was diagnosed as having a catheter-related bloodstream infection (CRBSI). His condition improved after two weeks of antifungal treatment. 

A 12-year-old boy had been admitted due to subarachnoid hemorrhage (SAH), which required a decompressive craniectomy. He had been diagnosed as having autism. On admission day 40, he had a high fever, and we removed the CVC and obtained two sets of blood cultures. Aerobic and anaerobic samples for the tip and two sets of blood culture yielded a yeast identified as *L. elongisporus* by the VITEK-2. Because of these results, he was diagnosed as having a CRBSI. He became afebrile as soon as the CVC was removed and the MCFG was started. The antifungal therapy had been performed for 2 weeks after the removal of the CVC. Consequently, his condition improved. Both sets of blood culture identified that the yeasts were susceptible to MCFG, as shown in Table 1. 

An 82-year-old woman had a medical history of diabetes mellitus, Parkinson’s disease and cholelithiasis. She was admitted due to cholangitis, which was successfully treated by endoscopic gallbladder stenting (EGBS) and drainage with antibiotic therapy by meropenem. She had a high fever, which was attributable a yeast infection revealed by the two sets of blood cultures as *L. elongisporus* by the VITEK-2 on day 28. Antifungal therapy with fosfluconazole (F-FLCZ) was started. However, after rechecking the blood cultures seven days after starting F-FLCZ, they continued to show positive results for *L. elongisporus*. Thus, F-FLCZ was switched to CPFG on day 39. She improved after 2 weeks of antifungal therapy with CPFG.

On CHROMagarTM candida (CHROMagar, Paris, France), there were turquoise blue colonies in all three cases, which looked different from *C. albicans* and *C. parapsilosis*, as shown in Figure 1A. Furthermore, a Gram-staining of *L. elongisporus* from an aerobic blood culture sample exhibited ellipsoidal to elongated ascospores (Figure 1B). The results of Basic Local Alignment Search Tool (BLAST) analysis showed the internal transcribe spacer (ITS)1 and ITS2 were 100% homologous with those of the *L. elongisporus* strain (CBS 2605) from GenBank on all three isolates. 

In total, 11 cases of *L. elongisporus* fungemia, including ours, have been previously reported, as shown in Table 1. The age of the patients ranges from 9 days to 82 years. Seven patients (64%) were male. Eight patients (73%) had some comorbidities such as diabetes mellitus or lung cancer. The remaining three patients had specific conditions, such as extremely low birth weight, autism, or postoperative trauma. Regarding infection sites, 4 of the 11 patients (36%) had CVCs and 5 patients were diagnosed as having CRBSI. As for the outcomes, 8 of the 11 patients (73%) were treated with echinocandins. Finally, six survived (55%), and five died (45%). Antifungal susceptibility testing was performed according to the Clinical and Laboratory Standards Institute (CLSI M27-A3). 

The minimum inhibitory concentrations (MICs) of 5-flucytosine (5-FC), amphotericin B (AMPH-B), fluconazole (FLCZ), MCFG and caspofungin (CPFG) are shown in Table 2. 

## 3. Discussion

Candidemia has emerged as an important nosocomial infection, showing a high mortality rate of 30–50%. “It is the fourth most common nosocomial bloodstream infection (BSI) in the United States, and the seventh most common nosocomial BSI in Europe and Japan. Previously reported risk factors for candidemia include central venous catheterization (CVC), neutropenia, malignancy, abdominal surgery within the previous 30 days, immunosuppressant use and admission to an intensive care unit (ICU)”. However, the etiology and the clinical features of *L. elongisporus* fungemia are still unknown. Although few *L. elongisporus* fungemia cases have been reported, they might have been misdiagnosed as *C. parapsilosis* fungemia. The widespread use of MALDI-TOF-MS makes it possible to diagnose *L. elongisporus* fungemia correctly, and the number of cases is estimated to increase. The etiology and the clinical features of this fungemia are still unknown. We found all the cases had either external devices or immunocompromised conditions. Thus, the risk of *L. elongisporus* fungemia might be similar and overlap with that of candidemia. The current Infectious Diseases Society of America (IDSA) guidelines favor the therapeutic use of echinocandins to treat *C. parapsilosis* [17,18]. Since there are no evidence-based breakpoints for antifungal susceptibility testing for *L. elongisporus*, those for *C. parapsilosis* are often used as an alternative [9,19]. In this review, five of the seven patients who received initial antifungal therapy with MCFG survived, including ours. Furthermore, all microorganisms isolated in our study showed that the MIC values for MCFG were ≤0.06, which were interpreted as susceptible, as shown in Table 2. Despite starting F-FLCZ, case 3 had persistent fungemia, shown by the positive blood cultures of *L. elongisporus*. Finally, switching to CPFG improved fungemia. Unfortunately, we could not find any reason for persistent fungemia while receiving F-FLCZ. Echinocandin might be the first choice for *L. elongisporus* fungemia.

As for a predictive tool for fungemia, we previously reported that combined sequential organ failure assessment (SOFA) and Charlson Comorbidity Index (CCI) scores could predict the outcomes among candidemia patients [4]. Multiple organ failure conditions and comorbidities contribute to a poor prognosis among candidemia patients. In our reports, all three cases displayed the combined SOFA (scores: 0–2) and CCI (scores: 1–3) scores of 2-3 on diagnosis as the fungemia, which were considered low, and eventually, all survived. This suggested that the combined score can be useful to predict the outcome of this fungemia, although the disease severity of the cases could be just not severe. We need more cases to evaluate whether this prognostic tool is helpful in the treatment of *L. elongisporus* fungemia. 

All patients with *L. elongisporus* fungemia had immunocompromised conditions and/or an external device. Thus, we hypothesize that these risk factors for *L. elongisporus* fungemia overlap with those of candidemia. More cases need to be accumulated to clarify the clinical features and epidemiology of *L. elongisporus* fungemia. Moreover, the clonal relationship of the three strains of *L. elongisporus* isolated should be analyzed for the prevention of nosocomial outbreaks in further analysis.

“Biofilm formation of *C. albicans* and *C. parapsilosis* in catheter or other implanted de vices in patients with weakened immune host defenses most often leads to the development of candidemia with serious repercussions for the health of these patients [20]. As early catheter removal may not always improve outcome, an effective antibiofilm treatment is of great clinical importance.” Echinocandins previously demonstrated a potent antibiofilm. Echinocandins previously demonstrated a potent antibiofilm formation activity in *C. albicans* and *C. parapsilosis* both in vitro and vivo studies [21]. *L. elongiporus* may have a potential to form a biofilm same as *C. albicans* and *C. parapsilosis*. Thus, *L. elongiporus* fungemia might have been persistent, even though antifungal agent therapy with F-FLCZ started in in the case 11. CPFG, is a fungicidal, water-soluble semisynthetic echinocandin that inhibits synthesis of β-1,3-d-glucan, a main structural component of the fungal cell wall, showing an antibiofilm formation activity. In a vitro study, Simitsopoulou et al. demonstrated that CPFG had a strong antibiofilm activity against *C. albicans*. This antifungal agents can be a useful treatment option for the treatment of *L. elongiporus* fungemia [22].

In conclusion, we experienced three successfully treated cases of *L. elongisporus* fungemia by MCFG. The risk factors of *L. elongisporus* fungemia may overlap with those of candidemia, and echinocandin can be a useful antifungal treatment for *L. elongisporus* fungemia.

## Figures and Tables

**Figure 1 microorganisms-11-01076-f001:**
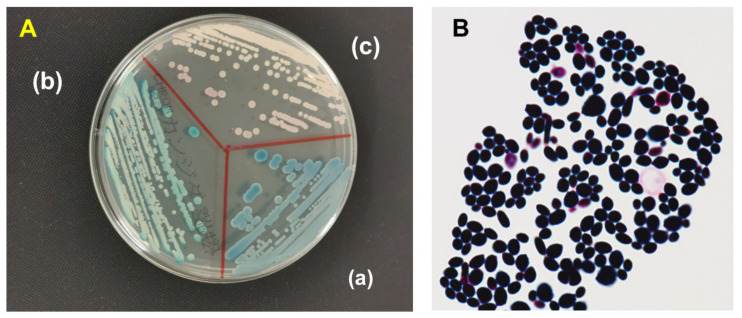
The plate was incubated at 35 °C for 48 h, (**A**) Colony characteristics of Lodderomyces elongisporus (a), *Candida parapsilosis* (b) and *C. albicans* (c) on CHROMagar Candia (CHROMagar, Paris, France). (**B**) Gram-stain of *L. elongisporus* from aerobic blood culture bottle. Gram-positive budding yeast were seen (magnification ×100).

**Table 1 microorganisms-11-01076-t001:** Cases of Lodderomyces elongisporus fungaemia previously published.

Case	Author (Year)	Age	Sex	Site of Infection	Underlying Diseases	External Device	Treatment	Outcome
1	Daveson [6] (2012)	30	M	Infectious endocarditis	Endocarditis, osteomyelitis and brain embolic lesions; intravenous drug user	None	CPFG→AMB	Survival
2	Ahmad [9] (2013)	63	M	CRBSI	Cardiovascular disease	None	FLCZ	Died
3	Taj-Aldeen [11] (2014)	22	M	Unknown	Trauma	Not described	CPFG	Survival
4	Hatanaka [10] (2016)	39	M	CRBSI	Thoracoabdominal aortic replacement complicated with aortoesophageal fistula	CVC	MCFG	Survival
5	Fernández-Ruiz [13] (2017)	79	M	Unknown	COPD, DM	None	CPFG	Died
6	Lee [12] (2018)	56	F	CRBSI	Lung cancer	CVC	Not prescribed	Died
7	Al-Obaid K [14] (2018)	71	F	Unknown	Ischemic heart diseases; peripheral vascular diseases	None	CPFG	Died
8	Asadzadeh [15] (2022)	9 days	F	Unknown	Extremely low-birth weight	Umbilical arterial and venous catheters	L-AMB	Died
9	Asai (2022)	76	M	CRBSI	DM	CVC	MCFG	Survival
10	Asai (2022)	12	M	CRBSI	Autism	CVC	MCFG	Survival
11	Asai (2022)	82	F	Unknown	DM, brain infarction, Parkinson’s disease, Cholelithic cholangitis	PV	F-FLCZ→CPFG	Survival

AMB, amphotericin B; CPFG, caspofungin; CRBSI, catheter-related bloodstream infection; CVC, central venous catheter; DM, diabetes mellitus; F, female; FLCZ, fluconazole; F-FLCZ, fosfluconazole; L-AMB, liposomal amphotericin B; M, male; PV, peripheral venous catheter.

**Table 2 microorganisms-11-01076-t002:** Antifungal susceptibility profile of *L. elongisporus* isolates.

Case	Author (Year)	Methods	MIC (ug/mL)					
			5-FC	AMPH-B	FLCZ	VRCZ	MCFG	CPFG
1	Daveson [6] (2012)	ND	0.06	0.25	≤0.125	≤0.008	ND	0.03
2	Ahmad [9] (2013)	Etest	0.094	ND	0.32	0.002	ND	0.094
3	Taj-Aldeen [11] (2014)	BMD, CLSI	ND	0.5	0.25	<0.016	ND	0.5
4	Hatanaka [10] (2016)	BMD, CLSI	0.5	0.25	0.5	0.015	0.015	ND
5	Fernández-Ruiz [13] (2017)	BMD, CLSI	ND	0.031	0.125	0.0017	0.015	ND
6	Lee [12] (2018)	ATB Fungus 3	1.00	0.25	1.00	0.12	0.06	0.25
7	Al-Obaid K [14] (2018)	Etest	0.064	0.012	0.125	0.004	0.032	0.064
8	Asadzadeh [15] (2022)	BMD	≤0.06	0.5	0.25	≤0.008	0.02	0.03
9	Asai (2022)	BMD, CLSI	≤1	≤0.25	≤0.25	≤0.12	≤0.06	≤0.12
10	Asai (2022)	BMD, CLSI	0.5	0.25	0.25	0.015	0.06	NE
11	Asai (2022)	BMD, CLSI	≤1	≤0.25	≤0.5	≤0.12	≤0.06	≤0.12

AMPH-B, amphotericin B; BMD, broth microdilution; CLSI, Clinical and Laboratory Standard Institute; CPFG, caspofungin; FLCZ, fluconazole; MCFG, micafungin; MIC, minimum inhibitory concentration; VRCZ, voriconazole; 5-FC, 5-Fluorocytosine.

## Data Availability

All authors meet the International Committee of Medical Journal Editors (ICMJE) authorship criteria for this article.

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
