# Peer review of "Three Successfully Treated Cases of Lodderomyces elongisporus Fungemia: Case Reports and a Review of the Literature"

_microorganisms, 2023, doi:10.3390/microorganisms11041076_

Round 1
Reviewer 1 Report
The study authors describe three cases of fungemia with Lodderomyces elongisporus, an emerging yeast infection and report on eight other cases in the literature. Overall, the case report and subsequent review provides a brief yet comprehensive overview of this emerging fungal infection and its potential for fungemia. However, certain points need to be addressed in relation to the details of the patient histories.
Major Comments
1. Patient #1 – the patient was reported to have uncontrolled DM. What was the latest HbA1c for him, if available? Additionally, was the mycotic inguinal artery aneurysm thought to be related to the fungal infection or a different cause? And what treatment did he get for it? That data would be important, as it would portend two risk factors for fungemia (poorly controlled DM and recent antibiotic use).
2. Patient #2 – the patient had catheter tip cultures positive for the fungus. What was the result of the blood cultures? How long did he receive antifungal therapy?
3. In Table 1 – the study authors have not included all the details of their particular cases in Patient 11. Additionally, they have put the year of publication as “2020”. Please add the relevant details and make the appropriate corrections.
4. The study authors have quoted the study by Lockhart et al (DOI: 10.1128/JCM.01790-07), that encountered 10 cases of Lodderomyces elongisporus fungemia that were mischaracterized as Candida parapsilosis. Lockhart et al did perform antifungal susceptibility testing, but there is no information on the clinical history of these patients. Was there susceptibility testing done on the other isolates of Lodderomyces elongisporus reported in the literature? It would be helpful to add those details to Table 2.
Minor Comments
1. In the Abstract, the study authors note that the fungus was identified in 1994, but then note in the Introduction that it was identified in 2008 as a causative human pathogen. Do they mean to say that it was morphologically identified in 1994, followed by isolation in humans in 2008? Please clarify this point
2. In the Introduction, the study authors mention that they “present two cases” but go on to describe three cases. Please rectify accordingly.
Author Response
Response to reviewer 1
Manuscript ID: microorganisms-2206828
Type: Case report
Title: Three successful treated cases of Lodderomyces elongisporus fungemia: Case reports and a review of the literature.
Thank you very much for reviewing my article “Three successful treated cases of Lodderomyces elongisporus fungemia: Case reports and a review of the literature.” I totally agree with your opinions and have revised the article according to your suggestions. The revised portions are in red in the text. I greatly appreciated your feedback and am positive that the revised article would be worth publication. Please, reconsider this article for publication in your journal.
Sincerely,
NOBUHIRO ASAI, and Hiroshige Mikamo on March 8, 2023
Major Comments
- Patient #1 – the patient was reported to have uncontrolled DM. What was the latest HbA1c for him, if available? Additionally, was the mycotic inguinal artery aneurysm thought to be related to the fungal infection or a different cause? And what treatment did he get for it? That data would be important, as it would portend two risk factors for fungemia (poorly controlled DM and recent antibiotic use).
A). Yes, I do agree with your opinion. That information is essential. Then, I added the following sentences.
In page 5
Case presentation
After a cardiac catheter test, a 76-year-old man was admitted to our institute due to an infectious aneurysm in the right inguinal artery. He had a history of poorly controlled diabetes mellitus with HbA1c 8.6% and angina pectoris. Since the causative pathogen for the disease was staphylococcus aureus, he received combination antibiotic therapy with sulbactam/ampicillin and daptomycin for 4 weeks. During the stay, he had a high fever, and two sets of blood cultures revealed yeast in both aerobic and anaerobic samples (BD, Japan).
- Patient #2 – the patient had catheter tip cultures positive for the fungus. What was the result of the blood cultures? How long did he receive antifungal therapy?
A). I am so sorry that I forgot to write the information down. So, I added some words and sentences as follows.
In page 5
A 12-year-old boy had been admitted due to subarachnoid hemorrhage (SAH) which required a decompressive craniectomy. He had been diagnosed as having autism. On admission day 40, he had a high fever, and we removed the CVC and obtained two sets of blood cultures. Aerobic and anaerobic samples for the tip and two sets of blood culture yielded a yeast identified as L. elongisporus by the Vitek 2. Because of these results, he was diagnosed as having CRBSI by L.elongisporus. He became afebrile as soon as the CVC was removed and the empirical therapy of MCFG was started. The antifungal therapy had been performed for 2 weeks after removal of CVC. Consequently, his condition improved. Both identified that the yeasts were susceptible to MCFG, as shown in Table 1.
- In Table 1 – the study authors have not included all the details of their particular cases in Patient 11. Additionally, they have put the year of publication as “2020”. Please add the relevant details and make the appropriate corrections.
A). I apologize for sending wring Table 1 which didn’t include detail information about the patient 11. I revised the table 1 appropriately.
- The study authors have quoted the study by Lockhart et al (DOI: 10.1128/JCM.01790-07), that encountered 10 cases of Lodderomyces elongisporusfungemia that were mischaracterized as Candida parapsilosis. Lockhart et al did perform antifungal susceptibility testing, but there is no information on the clinical history of these patients. Was there susceptibility testing done on the other isolates of Lodderomyces elongisporus reported in the literature? It would be helpful to add those details to Table 2.
A). I agree with your opinion. Then, I revised the table 2 including susceptibility testing in all isolates we could get. Please, check them and reconsider this article for publication.
Minor Comments
- In the Abstract, the study authors note that the fungus was identified in 1994, but then note in the Introduction that it was identified in 2008 as a causative human pathogen. Do they mean to say that it was morphologically identified in 1994, followed by isolation in humans in 2008? Please clarify this point
A). Thank you so much for your assist which made the article much better than before. You are right. The pathogen was identified first morphologically in 1994, followed by isolation as causative pathogen in human in 2008. According to your suggestion, I rewrote the sentence as follows.
In page 3
Abstract
Fungemia is a fatal systemic infection that can occur in immunocompromised patients. Despite that, antifungal stewardship is spreading widely but the mortality rate is extremely high, showing 40-60%. Loderomyces elongiporus is a newly detected pathogen morphologically, first described in 1994, followed by isolation in human in 2008. It has been misrecognized as candida parapsilosis. Recently, fever attributable to L. elongisporus fungemia cases have been reported, and the etiology and clinical features are still unknown. Here we present three successfully treated L. elongisporus fungemia cases by echinocandin.
- In the Introduction, the study authors mention that they “present two cases” but go on to describe three cases. Please rectify accordingly.
A). It`s my faut. Three is right. Then, I revied the sentence as follows.
In page 4
Lodderomyces elongisporus is an uncommon yeast opportunistic pathogen that can cause a sporadic fungal infection. This fungus was identified first in 2008 as a causative pathogen of human infection [6]. While several fungemia cases caused by this fungus have been reported [7-15], we still know little about its epidemiology, clinical features, risk or prognostic factors. Here, we present three cases of catheter-related bloodstream infection (CRBSI)s caused by L. elongisporus, which were successfully treated. Furthermore, we review all known reports published previously of fungemia caused by L. elongisporus.
Reviewer 2 Report
The work of Asai and cols. is a description of three cases of fungemia due to Lodderomyces elongisporus successfully treated with an equinocandin, as well as a literature review. In general, the case presentations along with the review and discussion are precise and concise. Moreover, the manuscript is adequately structured and moderately well written.
Just one recommendation, It would be highly desirable that the clonal relationship of the 3 strains of L. elongisporus isolated from the cases presented be analyzed, since it could perhaps be a nosocomial outbreak.
Author Response
Response to reviewer 2
Manuscript ID: microorganisms-2206828
Type: Case report
Title: Three successful treated cases of Lodderomyces elongisporus fungemia: Case reports and a review of the literature
Thank you very much for reviewing my article “Three successful treated cases of Lodderomyces elongisporus fungemia: Case reports and a review of the literature.” I totally agree with your opinions and have revised the article according to your suggestions. The revised portions are in red in the text. I greatly appreciated your feedback and am positive that the revised article would be worth publication. Please, reconsider this article for publication in your journal.
Sincerely,
NOBUHIRO ASAI, and Hiroshige Mikamo on March 3, 2023
Comments and Suggestions for Authors
The work of Asai and cols. is a description of three cases of fungemia due to Lodderomyces elongisporus successfully treated with an equinocandin, as well as a literature review. In general, the case presentations along with the review and discussion are precise and concise. Moreover, the manuscript is adequately structured and moderately well written.
Just one recommendation, It would be highly desirable that the clonal relationship of the 3 strains of L. elongisporus isolated from the cases presented be analyzed, since it could perhaps be a nosocomial outbreak.
A). Thank you so much for your great suggestion. I agree with your opinion that the clonal relationship of the 3 strains of L. elongisporus isolated from the cases presented be analyzed for a prevention of nosocomial outbreak. However, there is an issue of funding to do. Also, it will take a longer time to analyze them. Then, I added the sentence as follows.
In page 13 before conclusion
All patients with L. elongisporus fungemia had immunocompromised conditions and/or an external device. Thus, we hypothesize that these risk factors for L. elongisporus fungemia overlap those of candidemia. More cases need to be accumulated and clarify the clinical features and epidemiology of L. elongisporus fungemia. Moreover, the clonal relationship of the three strains of L. elongisporus isolated should be analyzed for the prevention of nosocomial outbreaks in further analysis.
In conclusion, we experienced three successfully treated cases of L. elongisporus fungemia by MCFG. The risk factors of L. elongisporus fungemia may overlap with those of candidemia, and echinocandin can be a useful antifungal treatment for L. elongisporus fungemia.
Round 2
Reviewer 1 Report
The study authors have incorporated most of the suggested changes in the manuscript and have helped increase the clarity of their case presentations. There are a few comments as below, particularly regarding the use of abbreviations, that the study authors should pay attention to before final publication.
Minor Comments
1. In the Abstract (line 31), please rectify “candida parapsilosis” to “Candida parapsilosis”.
2. The study authors have included the part of morphologic identification in 1994 in the Abstract but not in the Introduction. They should also include the relevant reference there.
3. Page 6 of 16, line 83 – the study authors use the abbreviation F-FLCZ. I presume this relates to Fluconazole, but this has not been expanded on. Please provide the full-form of the abbreviation
4. Adding to the point above, Fluconazole is abbreviated as FLCZ in the latter part of the manuscript (including in Tables 1 and 2). Please keep standardized abbreviations in the manuscript for ease of reading.
5. Table 2 – if using Fluconazole as an abbreviation FLCZ, it would be worthwhile to substitute that for the antifungal treatment for Case 2 as well.
6. Micafungin is abbreviated in the body of the manuscript but not in Table 1. It is important to maintain uniformity in the abbreviations. Similarly, Caspofungin has been abbreviated in the text as CPFG but not in Table 1.
7. Page 11 of 16, line 125 – the study authors have used “MULTI-TOF-MS” instead of what I presume was meant to be MALDI-TOF-MS. Please rectify.
8. Page 11 of 16, lines 129-130 – the study authors cite the IDSA guidelines in the line “The current Infectious Diseases Society of America (IDSA) guidelines favor the therapeutic use of echinocandins to treat C. parapsilosis”. Please provide the appropriate reference.
9. Pages 12 of 16 and 16 of 16 seem to have incorrect formatting, with part of the text being cut off. The study authors should rectify this.
Author Response
Response to reviewer 1
Manuscript ID: microorganisms-2206828
Type: Case report
Title: Three successful treated cases of Lodderomyces elongisporus fungemia: Case reports and a review of the literature.
Thank you very much for reviewing my article “Three successful treated cases of Lodderomyces elongisporus fungemia: Case reports and a review of the literature.” I totally agree with your opinions and have revised the article according to your suggestions. The revised portions are in red in the text. I greatly appreciated your feedback and am positive that the revised article would be worth publication. Please, reconsider this article for publication in your journal.
Sincerely,
NOBUHIRO ASAI, and Hiroshige Mikamo on March 12, 2023
Comments and Suggestions for Authors
The study authors have incorporated most of the suggested changes in the manuscript and have helped increase the clarity of their case presentations. There are a few comments as below, particularly regarding the use of abbreviations, that the study authors should pay attention to before final publication.
Minor Comments
- In the Abstract (line 31), please rectify “candida parapsilosis” to “Candida parapsilosis”.
A). I rewrote it according to your suggestion.
- The study authors have included the part of morphologic identification in 1994 in the Abstract but not in the Introduction. They should also include the relevant reference there.
A). Yes, it is necessary to add the citation. Then, I added the citation [6] as follows.
In page 4
Lodderomyces elongisporus is an uncommon yeast opportunistic pathogen that can cause a sporadic fungal infection. This fungus was newly identified first morphologically in 1994 [6], followed by isolation in 2008 as a causative pathogen of human infection [7].
- Page 6 of 16, line 83 – the study authors use the abbreviation F-FLCZ. I presume this relates to Fluconazole, but this has not been expanded on. Please provide the full-form of the abbreviation
A). I forgot to add full spelling on it. I revised it as follows.
In page 6
Antifungal therapy with fosfluconazole (F-FLCZ) was started. However, rechecking the blood culture seven days after starting F-FLCZ continued to show positive results for L. elongisporus. Thus, F-FLCZ was switched to CPFG on day 39. She improved after 2 weeks of antifungal therapy with CPFG.
- Adding to the point above, Fluconazole is abbreviated as FLCZ in the latter part of the manuscript (including in Tables 1 and 2). Please keep standardized abbreviations in the manuscript for ease of reading.
A). Both tables 1 and 2 were revised. I added all the words’ abbreviations.
- Table 2 – if using Fluconazole as an abbreviation FLCZ, it would be worthwhile to substitute that for the antifungal treatment for Case 2 as well.
A). As you mentioned, it is necessary to add it. Then, I added the following words “FLCZ, fluconazole;” below the table 1.
- Micafungin is abbreviated in the body of the manuscript but not in Table 1. It is important to maintain uniformity in the abbreviations. Similarly, Caspofungin has been abbreviated in the text as CPFG but not in Table 1.
A). Thank you for your suggestion. The table 1 was revised appropriately.
- Page 11 of 16, line 125 – the study authors have used “MULTI-TOF-MS” instead of what I presume was meant to be MALDI-TOF-MS. Please rectify.
A). I rerevised it as follows.
In page 11
Discussion
Although few L. elongisporus fungemia cases have been reported, they might have been misdiagnosed as C. parapsilosis fungemia. The widespread use of MALDI-TOF-MS makes it possible to diagnose correctly as L. elongisporus fungemia, and the number of cases is estimated to increase.
- Page 11 of 16, lines 129-130 – the study authors cite the IDSA guidelines in the line “The current Infectious Diseases Society of America (IDSA) guidelines favor the therapeutic use of echinocandins to treat C. parapsilosis”. Please provide the appropriate reference.
A). Sorry about that the ciations were missing. I added the citations as follows.
In page 11
The current Infectious Diseases Society of America (IDSA) guidelines favor the therapeutic use of echinocandins to treat C. parapsilosis [17,18]
- Pages 12 of 16 and 16 of 16 seem to have incorrect formatting, with part of the text being cut off. The study authors should rectify this.
A). Thank you for your suggestion. I cut the blank page. Let me know if it needs to be fixed more.
Reviewer 2 Report
The authors have made the pertinent explanatory notes in the revised version of the manuscript. I recommend its acceptance.
Author Response
Response to reviewer 2
Manuscript ID: microorganisms-2206828
Type: Case report
Title: Three successful treated cases of Lodderomyces elongisporus fungemia: Case reports and a review of the literature.
Thank you very much for reviewing my article “Three successful treated cases of Lodderomyces elongisporus fungemia: Case reports and a review of the literature.” I totally agree with your opinions and have revised the article according to your suggestions. The revised portions are in red in the text. I greatly appreciated your feedback and am positive that the revised article would be worth publication. Please, reconsider this article for publication in your journal.
Sincerely,
NOBUHIRO ASAI, and Hiroshige Mikamo on March 12, 2023
Comments and Suggestions for Authors
The authors have made the pertinent explanatory notes in the revised version of the manuscript. I recommend its acceptance.
A). Thank you so much for your excellent reviewing. I really appreciate your great help which made the article much nicer than previous one.
Best regards,
Nobuhiro Asai and Hiroshige Mikamo on March 12,2023